# Are digital social media campaigns the key to raise stroke awareness in low-and middle-income countries? A study of feasibility and cost-effectiveness in Nepal

Christine Tunkl[1]*, Raju Paudel[2], Lekhjung Thapa[3], Patrick Tunkl[4], Pankaj Jalan[5], Avinash Chandra[6], Sarah Belson[7], Bikram Prasad Gajurel[8], Nima Haji-Begli[1], Sunanjay Bajaj[1], Jessica Golenia[1], Wolfgang Wick[1,9], Werner Hacke[1], Christoph Gumbinger[1]

1 Department of Neurology, University Hospital Heidelberg, Heidelberg, Germany, 2 Grande International Hospital, Kathmandu, Nepal, 3 National Neuro Center, Kathmandu, Nepal, 4 Tunkl Consulting, Heidelberg, Germany, 5 Norvic International Hospital, Kathmandu, Nepal, 6 Annapurna Neurological Institute, Kathmandu, Nepal, 7 World Stroke Organization, Geneva, Switzerland, 8 Tribhuvan University Teaching Hospital, Kathmandu, Nepal, 9 German Cancer Research Center, Heidelberg, Germany

* christine.tunkl@med.uni-heidelberg.de

**Data Availability Statement:** The minimal dataset for this study is publicly available on https://figshare.com/articles/dataset/Original_social_

## Abstract

### Background

Stroke is a major global health problem and was the second leading cause of death world-wide in 2020. However, the lack of public stroke awareness especially in low- and middle-income countries (LMICs) such as Nepal severely hinders the effective provision of stroke care. Efficient and cost-effective strategies to raise stroke awareness in LMICs are still lacking. This study aims to (a) explore the feasibility of a social media-based stroke awareness campaign in Nepal using a cost-benefit analysis and (b) identify best practices for social media health education campaigns.

### Methods

We performed a stroke awareness campaign over a period of 6 months as part of a Stroke Project in Nepal on four social media platforms (Facebook, Instagram, Twitter, TikTok) with organic traffic and paid advertisements. Adapted material based on the World Stroke Day Campaign and specifically created videos for TikTok were used. Performance of the campaign was analyzed with established quantitative social media metrics (impressions, reach, engagement, costs).

### Results

Campaign posts were displayed 7.5 million times to users in Nepal. 2.5 million individual social media users in Nepal were exposed to the campaign on average three times, which equals 8.6% of Nepal's total population. Of those, 250,000 users actively engaged with the posts. Paid advertisement on Facebook and Instagram proved to be more effective in terms

media_data/22816361. Additional data from the Social Media Business Accounts are available from the University Hospital Heidelberg (contact via neurologie@med.uni-heidelberg.de) for researchers who meet the criteria for access to confidential data.

**Funding:** This study is supported by Hospital Partnerships funding program of the Deutsche Gesellschaft für internationale Zusammenarbeit (GIZ) GmbH and received funding by the Federal Ministry of Economic Cooperation and Development (BMZ) and the Else Kröner-Fresenius foundation (EKFS). The funders had no role in study design, data collection and analysis, decision to publish, or preparation of the manuscript.

**Competing interests:** The authors have declared that no competing interests exist.

of reach and cost than organic traffic. The total campaign cost was low with a "Cost to reach 1,000 users" of 0.24 EUR and a "Cost Per Click" of 0.01 EUR.

## Discussion

Social media-based campaigns using paid advertisement provide a feasible and, compared to classical mass medias, a very cost-effective approach to inform large parts of the population about stroke awareness in LMICs. Future research needs to further analyze the impact of social media campaigns on stroke knowledge.

## Introduction

Stroke is a major global health problem and is the second leading cause of death worldwide, with the highest burden of diseases in low-and middle-income countries (LMICs) [1]. Though the clinical outcome of strokes can be significantly improved by rapid diagnosis and treatment, patients repeatedly arrive late to hospitals [2] whereby the time-dependent effect of treatment decreases the chance of a good outcome [3]. The lack of public awareness about stroke has been reported as one of the major factors for prehospital delay and hence the belated medical treatment [4–6].

In Nepal, recent studies demonstrated that knowledge on stroke in high-school students and in a rural population is insufficient [7, 8] with e.g. 55% of study participants believing in ayurvedic therapy to be effective in stroke care. The lack of awareness regarding stroke treatments and inadequate recognition of stroke symptoms have been identified as main factors contributing to delayed presentation at a tertiary care center [9] and therefore hampering the benefit of treatment.

In 2020, the Nepal Stroke Association and University Hospital Heidelberg initiated a project aiming to improve stroke care in Nepal and acknowledging the need to focus on public awareness of stroke. The term 'stroke awareness' refers mainly to stroke recognition, specifically its symptoms and the need for the correct emergency response [10]. Given the limited resources in low- and middle income countries (LMICs) it is therefore crucial to develop an approach which 1) reaches a large group of the population, 2) leads to a short- and long-term increase in stroke awareness and 3) is cost-effective [11].

The effectiveness of classical mass educational campaigns on a community's stroke knowledge has proven to be low and does often not lead to a long-lasting change in help-seeking behavior [12, 13]. In the last decade, social media has emerged as a platform with an enormous outreach with more than five billion people worldwide owning a smartphone [14]. By facilitating information sharing opportunities and community-building, social media has become invaluable in marketing and a promising approach in health education [15]. Especially young adults have expressed their interest in receiving health information via social media platforms [16] and a review found that social media even has the ability to facilitate mass communication, health education and knowledge translation in LMICs. Therefore, newer innovative approaches have focused on children and young adults as the recipients of stroke education, assuming that those age groups will be the bystanders when a stroke happens and take the right help-seeking actions [17]. Furthermore, recent studies have shown that these groups will deliver the message to their family members [17–19]. As social media covers a large audience of over 50% of the population in Nepal [20], we hypothesized that a social media campaign in Nepal is feasible to reach out the Nepalese people, might be effective in terms of knowledge

acquisition by engaging the population and requires a lower financial input compared to classical mass media campaigns.

The objectives of this study were to (a) determine feasibility of a social media-based stroke awareness campaign in Nepal, (b) analyze the effectiveness of the campaign and (c) identify best practices for social media health education campaigns.

Addressing this question has the potential to provide innovative strategies for health education and could support the development of future interventions to increase awareness of stroke and other relevant health issues.

## Materials and methods

We performed a non-controlled cohort-study based on exposure to advertising of stroke awareness materials on social media targeting users in Nepal.

### Setting

Nepal is a LIMC with a population of approximately 29.1 million, and is ranked 143rd in the Human Development Index. The potential study population included the 13.7 million social media users in Nepal (equals 45.7% of the total population) [21]. We used Facebook, Instagram, Twitter and TikTok as at these platforms have the largest audience in Nepal [22]. Data published in the channel's advertising resource indicated that Facebook had 12.3 million users, Instagram 2.3 million users, TikTok 2.2 million users and Twitter 417,000 users in early 2022 [22].

The study was performed continuously over a period of 24 weeks between January and June 2022 as part of the "Nepal Stroke Project".

### Content dissemination and study population

We used two different methods of content dissemination, which was organic traffic (on all four platforms) and paid-for promoted posts on Facebook and Instagram (Meta Network). Organic social media refers to free content, permanently seen in the account's timeline, which is displayed to the account's followers, to follower's followers and to users who were "hash-tagged" or who searched specifically for a post's hashtag. Content of organic traffic on all four social media channels targeted network effects with initial dominance of Nepalese users and links to locations in Nepal aimed to disseminate the content mainly within the Nepalese population but was formally visible without any geographic restrictions.

Paid advertisements (ads) on Facebook and Instagram targeted users residing in Nepal over the age of 13, as specified by Meta Platforms Inc., without any further exclusion criteria. Ads containing creatives in English language were displayed to users who use social media in English and those in Nepali language to users with Nepali device settings. Paid-for promoted posts work on a "cost per click" (CPC) basis, whereby an allocated budget per post was set with the post being promoted to a specified target audience until the budget is reached.

The study size was determined by the output in terms of reach of a predefined action plan and a predefined budget limit for paid advertisement (for details see Table 1).

### Channels, content, informative website

The study was conducted using the Facebook and Instagram pages of the Nepal Stroke Project, the Twitter handle @nepalstrokeproject and the TikTok account @NepalStroke, which were all created in November 2021. An additional business account was used for Facebook and Instagram to enable paid-for advertisements and access data analytics not available for standard

**Table 1. Key performance indicators available per platform.**

| Metric | Facebook® | Instagram® | TikTok® | Twitter® | Paid-Ads (Meta®) |
|---|---|---|---|---|---|
| Impression[1] | No | No | Yes | Yes | Yes |
| Reach[2] | Yes | Yes | No | No | Yes |
| Engagement[3] | Yes | Yes | Yes | Yes | Yes |
| Link-Click[4] | No | No | No | No | Yes |
| Cost-Per-Click[5] | No | No | No | No | Yes |

[1] Impressions: Number of times a piece of content was displayed to a target audience.

[2] Reach: Number of users exposed to a piece of content.

[3] Engagement: Number of interactions the piece of content received from user, such as reactions, shares, comments, link clicks, 3-seconds video plays.

[4] Link click: The number of clicks on links within an ad that users clicked on.

[5] Cost per Click: The cost to pay for each click on an advertisement

user accounts. While Facebook and Instragram were used to promote the World Stroke Day Campaign, Twitter was used exclusively for promoting project-specific information.

## Content and creatives

We used the widely used FAST message (face, arms, speech, time), based on the Cincinnati Pre-Hospital Stroke Scale [23]. The FAST message is used in many countries worldwide and is the key message of the World Stroke Organization's (WSO) World Stroke Day (WSD) Campaign (Fig 1). The FAST acronym has been translated into Nepali language and has been pre-tested in educational campaigns of the Nepal Stroke Association to ensure applicability in Nepal. Materials used in the campaign were mainly based on the 2021 WSD Campaign from WSO, which combines the FAST message with appealing slogans (e.g.: *Minutes can save memories)* and the Call to Action: *Learn the signs*, *Say it's a stroke and save #precious time*. The used static and animated key visuals combine an emotional hook, call to action and symptom spotting. The key visuals are available with ten different images, covering different ethnicities (Fig 2) and additionally contain icons which are simple, genderless and without ethnicity (Fig 3).

All creatives have been translated into Nepali language and script by an expert team of Nepalese health workers and reviewed for linguistic and content correctness by professional translators. The creatives are supplied by own-designed posts with general facts on stroke and modifiable risk factors as well as posts to celebrate national festive events, inform about the project progress and building a community (Fig 4). All posts included a call to action to access the informative stroke website.

Special posts were designed for Twitter restricted to 160 characters and focusing on the text message. TikTok requires a different format with short videos of less than one minute, who were specially created by a professional TikTok creator in Nepali language and reviewed for content correctness by Nepalese stroke physicians.

## Informative website

A website to provide evidence-based and comprehensive stroke knowledge, called "The Basics of Stroke", was developed with the aim to explain basic stroke knowledge in plain English and Nepali language (i.e. risk factors, symptoms, treatment options). The content of the website was developed by a working group consisting of Nepalese and German stroke physicians, nurses and online marketing experts (https://nepalstrokeproject.org/the-basics-of-stroke/).

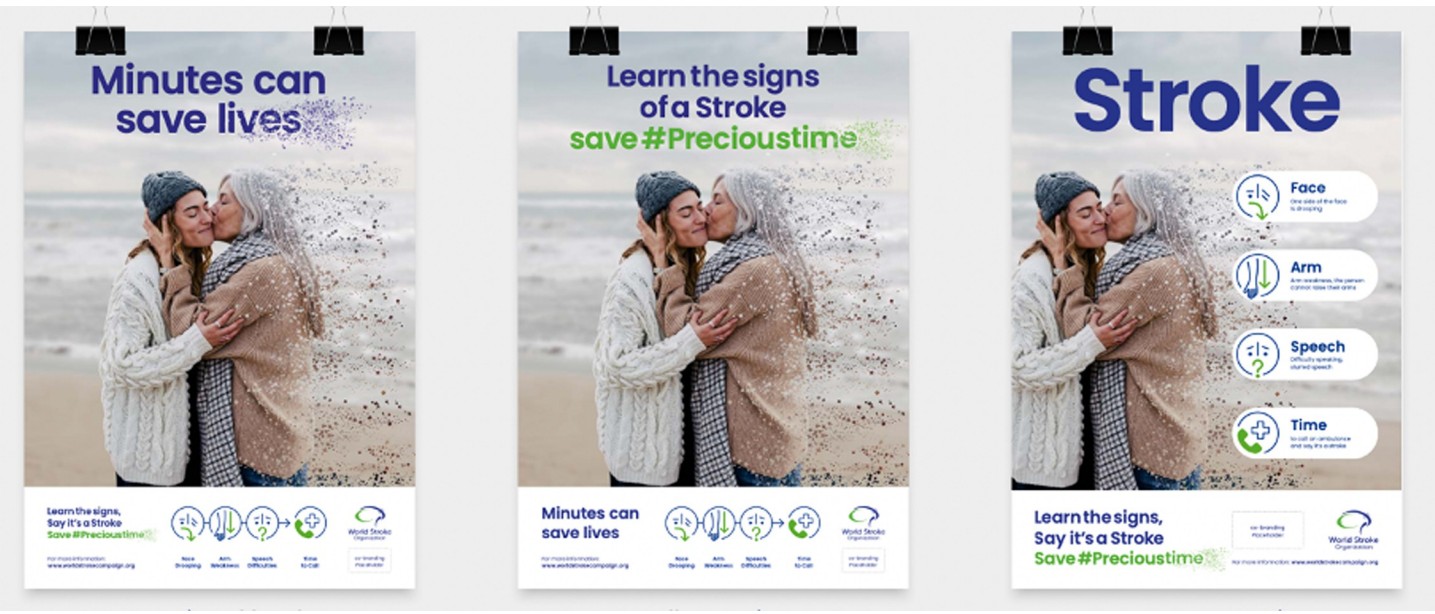

**Fig 1. Key visuals of social media campaign.** Fig 1 displays selected key visuals of the World Stroke Day Campaign. Republished from World Stroke Day Campaign under a CC BY license, with permission from World Stroke Organization, original copyright 2022.

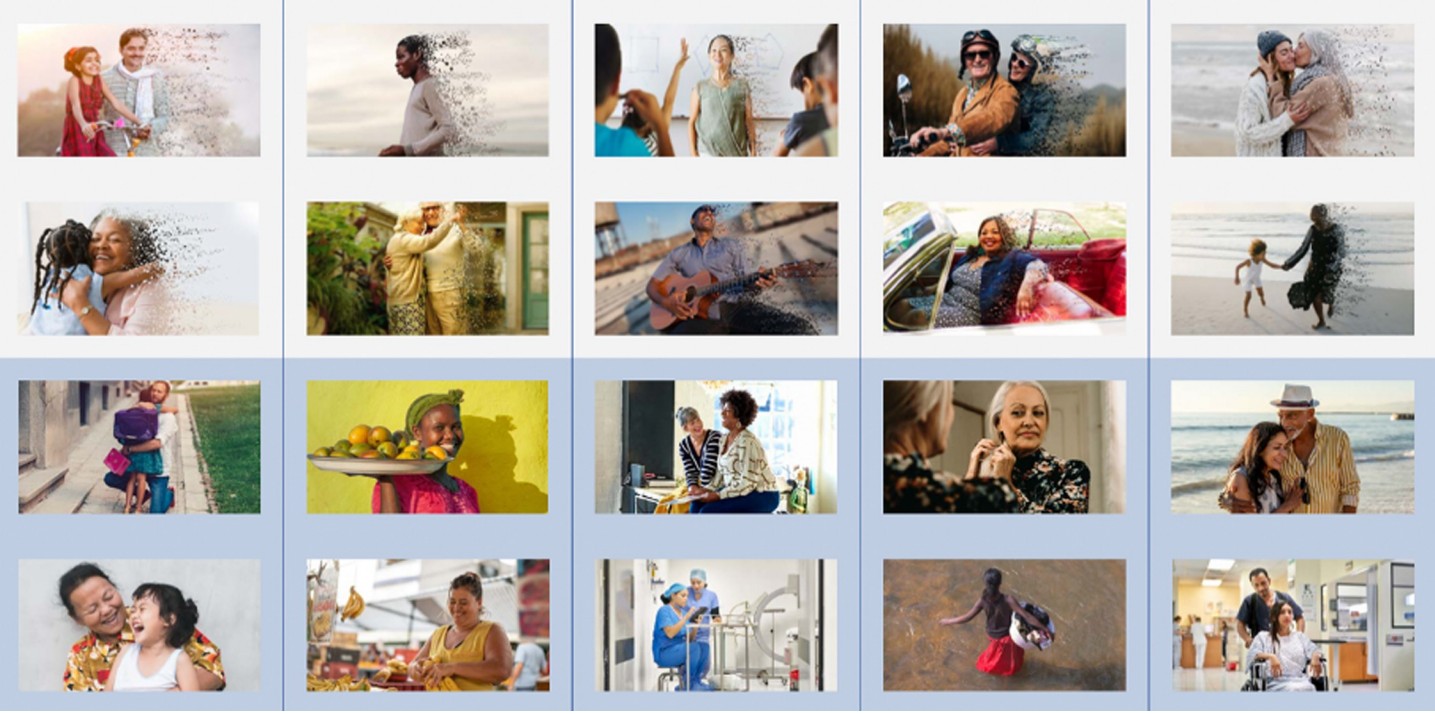

**Fig 2. Key images of social media campaign.** Fig 2 displays the key images of the World Stroke Day Campaign. Republished from World Stroke Day Campaign under a CC BY license, with permission from World Stroke Organization, original copyright 2022.

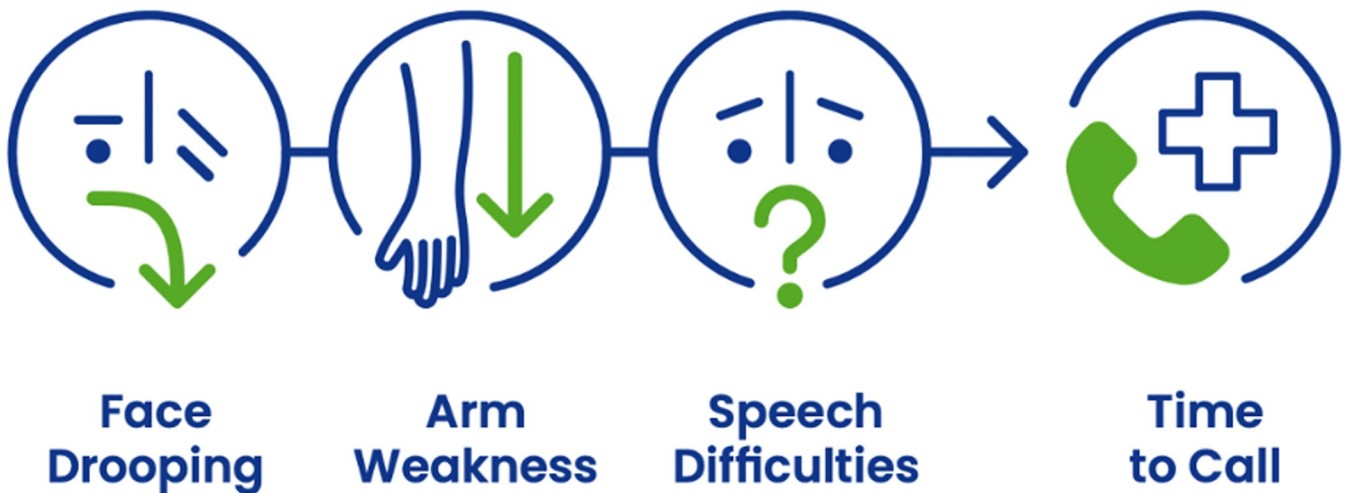

**Fig 3. Iconography of the FAST message.** Fig 3 displays the iconography of the World Stroke Day Campaign. Republished from World Stroke Day Campaign under a CC BY license, with permission from World Stroke Organization, original copyright 2022.

### Course of action

A predefined action plan combining organic traffic for Facebook, Instagram, Twitter and Tik-Tok and paid-for advertisements on Facebook and Instagram was developed by a marketing expert. For organic traffic it consisted of a less intensive run-in phase (month 1), a more intensive phase (month 2/3) with more posts per time-unit and a constant phase (starting in month 3). For cost minimization, paid advertisements were run at constant intervals. The editorial plan envisioned the following activities:

- Posts on Facebook, Instagram and Twitter to be distributed by organic traffic

- Two paid advertisements per month on Facebook and Instagram with each being promoted for seven to ten days including several days on social without advertisements.

- Two videos per week on TikTok starting in April 2022.

### Variables, data sources, data management

To assess the feasibility of the social media campaign we measured the following quantitative key performance indicators (KPI), which differ slightly between the platforms [24, 25] (Table 1):

- <u>Impressions</u>: The number of times a piece of content was displayed to a target audience

- <u>Reach</u>: The number of users exposed to a piece of content

- <u>Engagement</u>: The number of interactions the piece of content received from user, such as reactions, shares, comments, link clicks, 3-seconds video plays.

- <u>Engagement rate</u>: The ratio of engagements to impressions.

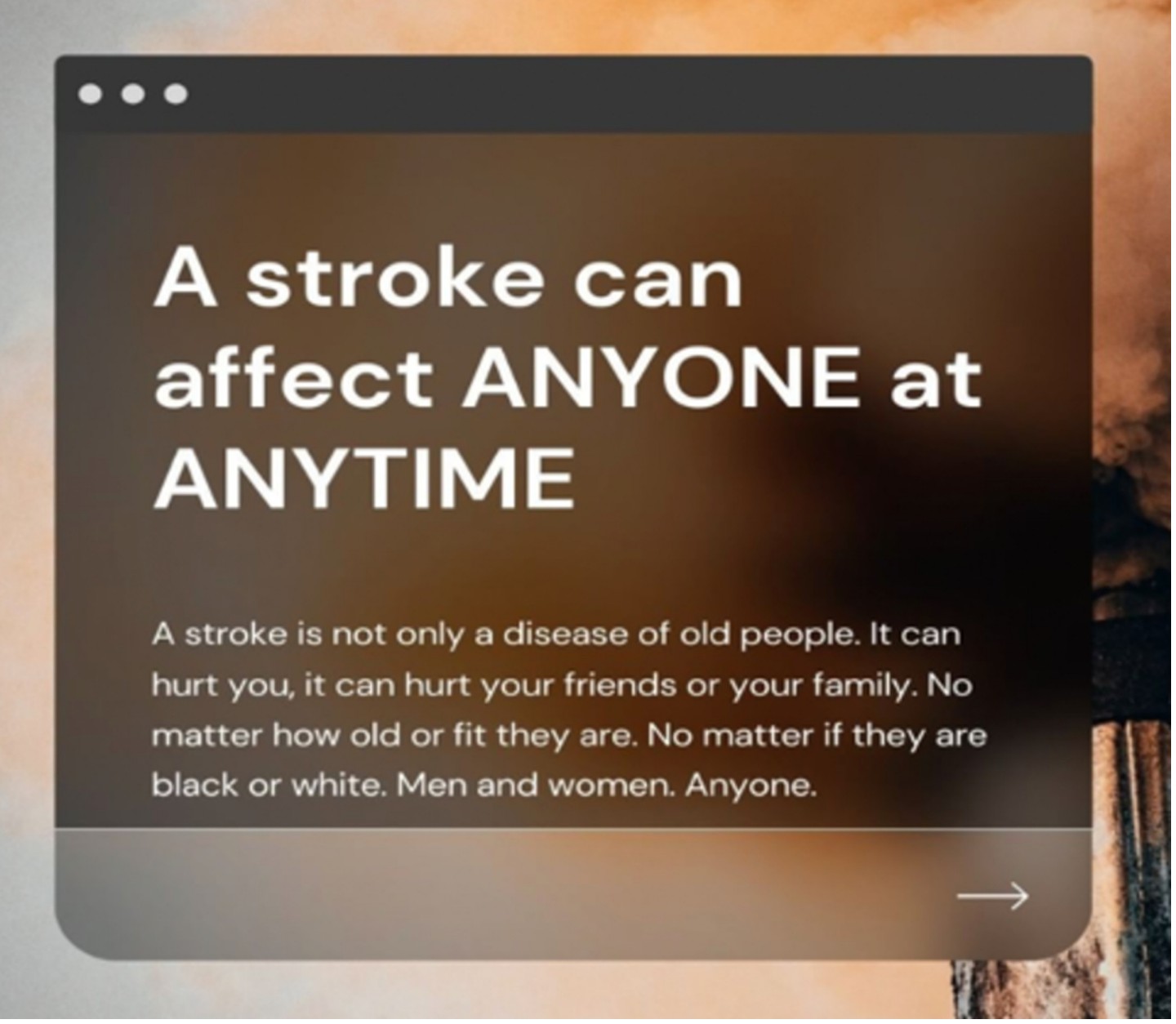

**Fig 4. Self-designed creatives with general stroke knowledge.** Fig 4 shows an example of a self-designed post used for social media posts.

- Link clicks: The number of clicks on links within an ad that users clicked on

- Cost per mille (CPM): The cost to pay for 1,000 impressions on a social media platform

- Cost per Click (CPC): The cost to pay for each click on an advertisement

   Campaign performance for platforms and individual material performance was monitored by extracting data from Meta Business Manager, Twitter Analytics and TikTok Account Manager.

   All data was entered into a database (Excel, Version 2019, Microsoft Corp, USA). Data was extracted at the end of the campaign (30/06/2022).

## Statistical analysis

Feasibility of the campaign was explored through descriptive statistics of cumulative reach, impressions and engagement over time as measured from monitoring data on all platforms. To evaluate the cost-effectiveness of the campaign we analyzed the Cost-per-Click (CPC), Cost per mille (CPM) and the cost of working hours and preparation of material.

Best practices on social media-based health education were identified by a comparison of reach and engagement rate on different modalities (paid vs organic traffic) and different channels.

## Ethical clearance

This study was approved by the Nepal Health Research Council (Reference number 66 / 703–2021 P). According to the terms of service and data policy of the Meta Platform, Twitter and TikTok all users have agreed into providing the analyzed information about e.g. hashtags, likes, views, time, frequency and duration of views [26–28]. As we used only aggregated user analytics we did not need to obtain an additional vote from our local ethic committee (as to our local standards).

# Results

Between January and June 2022, on average one post per week was broadcasted by organic traffic on Facebook, Instagram and Twitter, fourteen paid-for promoted posts ran on Facebook and Instagram and 28 short videos were posted on TikTok (see Table 2).

## Basic demographic information on study population

Basic demographic information was only collected in the users exposed to paid advertisement, which however account for 97.9% of the study population. All users were residents of Nepal as specified by the prior defined inclusion criteria. In accordance with the sex distribution on Facebook and Instagram in Nepal, 44% of study participants were female and 56% were male. 69% of all exposed users were under the age of 35 (see Table 3).

## Outreach

The social media campaign material was displayed more than seven million times (n = 7,513,952) on four different social media platforms (Facebook, TikTok, Instagram, Twitter) (Table 4).

The paid advertisements on Facebook and Instagram reached nearly 2.5 million unique users (n = 2,411,505) mainly in Nepal, equaling 8.6% of Nepal's total population or 17.9% of all social media users in Nepal, and each user was exposed to the campaign on average three times.

Additionally, organic traffic on all platforms was viewed by a maximum number of 161,174 users but due to an overlap between different platforms and potential multiple views per users, we did not include these (additionally relatively small number) views in further analyses.

In total over a quarter million people (n = 265,095) interacted with our content as measured by likes, shares, comments or link clicks (0.9% of Nepal's total population). The link click to the informative website as the most meaningful performance indicator [29] was achieved in 257,999 people in Nepal (Table 4).

**Table 2. Social media implementation plan.**

| Month | Jan | | | | Feb | | | | March | | | | April | | | | | May | | | | June | | | |
|---|---|---|---|---|---|---|---|---|---|---|---|---|---|---|---|---|---|---|---|---|---|---|---|---|---|
| Week | 1 | 2 | 3 | 4 | 5 | 6 | 7 | 8 | 9 | 10 | 11 | 12 | 13 | 14 | 15 | 16 | 17 | 18 | 19 | 20 | 21 | 22 | 23 | 24 | 25 |
| Facebook® | | 1 | 1 | | 2 | 1 | 2 | | 2 | | 1 | 1 | 1 | 1 | 1 | 1 | | 1 | 1 | 1 | 1 | 1 | 1 | 1 | 1 |
| Instagram® | | | 1 | | 1 | 1 | 1 | 2 | 2 | 1 | 1 | | | 1 | 1 | 1 | 1 | | 1 | 1 | 1 | 1 | 1 | 1 | 1 |
| Tiktok® | | | | | 2 | 1 | 2 | 2 | | 1 | 1 | 1 | 1 | 1 | | | 1 | | | | | 1 | | 1 | |
| Twitter® | | | | | | | | | | 1 | 1 | | | | | | | 2 | 2 | 2 | 2 | 2 | 2 | 2 | 2 |
| Paid-Ads[1] (Meta) | | | 1 | 1 | | 1 | | 1 | | 1 | 1 | 1 | | | 1 | 1 | | 1 | 1 | 1 | | 1 | 1 | 1 | |

[1] Paid Ads: Paid advertisement on social media platforms in the Meta® network. Numbers indicate number of posts / month.

## KPI of organic versus paid-for traffic in social media-based awareness campaigns

The overall performance was significantly better in paid-for advertisements compared to organic traffic, though the number of posts on organic traffic was 5-fold higher (n = 77: Facebook n = 23, Instagram n = 21, TikTok n = 18, Twitter n = 15) compared to paid advertisements (n = 14) (Table 4). Paid-for advertisement resulted in 45 times more impressions than organic traffic (n = 7,352,777 versus 161,085 impressions). From all impressions achieved on four social media platforms (n = 7,513,952), 97.9% were contributed to paid-for advertisements. While 13,993 unique users were exposed to organic posts on Facebook and Instagram streams, a 172-fold higher number of users was exposed to paid-for advertisement on the same channels (n = 2,411,505). We assumed that active engagement with a post is associated with more gain in knowledge than only being displayed a post [29] and the overall campaign performance showed an active engagement with any posts in 265,095 users with 97.3% attributable to paid ads.

The engagement rate (number of engagements/ number of impressions) did not show a relevant difference between organic and paid traffic (4.4% on organic vs 3.5% on paid traffic). Interestingly, when considering only Facebook and Instagram, organic traffic achieved a high

**Table 3. Basic demographics of users exposed to paid advertisements.**

| | Reach[1] (N) | Reach (%) | Link-clicks[2] (N) | Link-clicks (%) |
|---|---|---|---|---|
| | (n = 2,411,505) | | (n = 257,990) | |
| **Age** | | | | |
| 65+ | 127,999 | 5.31 | 13,866 | 5.4 |
| 55–64 | 91,647 | 3.80 | 10,318 | 4.0 |
| 45–54 | 150,015 | 6.20 | 17,981 | 7.0 |
| 35–44 | 373,246 | 15.50 | 40,237 | 15.6 |
| 25–34 | 532,989 | 22.10 | 53,512 | 20.8 |
| 18–24 | 524,797 | 21.80 | 36,792 | 14.3 |
| 13–17 | 610,812 | 25.30 | 85,284 | 33.6 |
| **Sex** | | | | |
| Female | 1,069,049 | 44.3 | 106,279 | 41.2 |
| Male | 1,339,896 | 55.6 | 151,590 | 58.8 |
| Non classified | 2,560 | 1.1 | 121 | 0.04 |

[1] Reach: The number of users exposed to a piece of content.

[2] Link click: The number of clicks on links within an ad that users clicked on.

Table 4. Key performance indicators of social media platforms.

| Metric | Facebook® | Instagram® | TikTok® | Twitter® | Paid-Ads (Meta®) |
|---|---|---|---|---|---|
| Impression[1] | 8,219 | 6,753 | 139,481 | 6,722 | 7,352,777 |
| Reach[2] | 7,665 | 6,328 | n.a. | n.a. | 2,411,505 |
| Engagement[3] | 790 | 190 | 5,738 | 378 | n.a. |
| Link-Click[4] | n.a. | n.a. | n.a. | n.a. | 257,999 |
| Cost-Per-Click[5] | n.a. | n.a. | n.a. | n.a. | 0.009 EUR |

[1] Impressions: Number of times a piece of content was displayed to a target audience.

[2] Reach: Number of users exposed to a piece of content.

[3] Engagement: Number of interactions the piece of content received from user, such as reactions, shares, comments, link clicks, 3-seconds video plays.

[4] Link click: The number of clicks on links within an ad that users clicked on.

[5] Cost per Click: The cost to pay for each click on an advertisement

engagement rate of 6.5% on these platforms. Considering only organic traffic on all four platforms, TikTok performed the best in terms of impressions (n = 139,481) compared to Facebook (n = 8,219), Instagram (n = 6,753) and Twitter (n = 6,722). In terms of engagement rates on all platforms with organic traffic, Facebook had the best rate (Facebook: 9.6%; Instagram: 2.8%; Twitter: 5.6%, TikTok 4.1%), though achieving only a small number in absolute numbers (n = 790 engagements).

## Cost analysis of organic and paid traffic

In total, we spent 3,474 EUR, reaching 8.6% of Nepal's total population and engaging over a quarter million of people in Nepal using a social media-based approach.

All posts for organic traffic were free of charge and the creatives by World Stroke Day Campaign material were provided free of charge. TikTok required its own short video format, the costs for each video production were 20 EUR, which adds to a total of 1,130 EUR and a CPM of 7 EUR.

The total pre-defined budget for paid advertisements on Facebook and Instagram was 2,000 EUR with additional cost of a marketing specialists' working time of 280 EUR for 14 posts (15 minutes time/ post). On average, 0.24 EUR were spent for each 1,000 impressions (cost-per-1,000 impressions, CPM) and 0.01 EUR was spent for each link-click to the informative website (CPC). Dissemination costs of advertisements were constant over time (see Table 5).

## Discussion

### Principal findings

Our cohort study demonstrates that a social media- based campaign with paid advertisement is feasible to reach and provoke reactions from a large proportion of the population in Nepal (reach: n = 2,425,498 users, i.e. 8.6% of the total population, engagement: n = 265,095 users, i.e. 0.9% of Nepal's total population) on a low budget. In particular, the CPC of 0.01 EUR highlights that paid advertisements on social media may be a cost-effective method for public health awareness campaigns.

### Feasibility of social-media based awareness campaigns

Our social media-based campaign on four different platforms (Facebook, Instagram, TikTok, Twitter) with additional paid advertisements on Facebook and Instagram exposed 8.6% of the

**Table 5. Key performance indicators of paid advertisement on meta network.**

| Campaign name | Language | Channel | Link-Clicks[1] | Reach[2] | Impression[3] | CPC[4] in EUR | Amount spent (EUR) |
|---|---|---|---|---|---|---|---|
| WSD[5] - AV[6]: Lives; Mobility | NP[7] | FB[8], IN[9] | 103,035 | 690,048 | 1,412,830 | 0.01 | 1,000 |
| WSD—AV: Speech, Memories | NP | FB, IN | 64,256 | 592,766 | 968,386 | 0.00 | 210 |
| WSD—AV: Speech, Memories | NP | FB, IN | 43,696 | 346,051 | 709,627 | 0.01 | 257.82 |
| WSD Animated FAST icons | NP | FB, IN | 23,747 | 726,509 | 1,404,574 | 0.01 | 209.99 |
| Own creative: Signs of stroke | EN[10] | FB, IN | 8,173 | 375,040 | 647,619 | 0.01 | 93.52 |
| Quiz (signs & symptoms) | EN | FB, IN | 3,804 | 176,192 | 295,187 | 0.01 | 46.11 |
| Quiz (signs & symptoms) | EN | FB, IN | 3,284 | 63,488 | 287,536 | 0.01 | 49.02 |
| Quiz (signs & symptoms) | EN | FB, IN | 2,811 | 58,430 | 116,763 | 0.01 | 40 |
| Quiz (signs & symptoms) | NP | FB, IN | 1,520 | 15,910 | 21,502 | 0.00 | 5.8 |
| Own creative: Facts on stroke | EN | IN | 1,146 | 173,790 | 368,892 | 0.09 | 104.99 |
| Quiz (signs & symptoms) | NP | FB, IN | 996 | 66,352 | 74,629 | 0.01 | 10.32 |
| Own creative: Facts on stroke | EN | IN | 613 | 255,168 | 675,789 | 0.18 | 112.91 |
| Quiz (facts on stroke) | EN | IN | 548 | 54,800 | 78,239 | 0.04 | 23.38 |
| Own creative: | EN | IN | 361 | 131,520 | 291,204 | 0.11 | 40.84 |
| project awareness | | | | | | | |
| **Total** | | | 257.990 | 2.411.505 | 7.352.777 | 0,0085 | 2204,7 |

Meta® paid advertisement report. Report period: Jan 1, 2022—Jun 30, 2022.

[1] Link click: The number of clicks on links within an ad that users clicked on.

[2] Reach: The number of users exposed to a piece of content.

[3] Impressions: The number of times a piece of content was displayed to a target audience.

[4] CPC: cost per click.

[5] WSD: World Stroke Day.

[6] AV: animated visuals.

[7] NP: Nepali.

[8] FB: Facebook®.

[9] IN: Instagram®.

[10] EN: English.

country's total population to stroke awareness material and achieved an active engagement of 250,000 people in Nepal, which equals 1% of the country's whole population.

An advantage of social media- based health campaigns is their ability to provide tailored messages to the target audience. The digital items provided by the World Stroke Day (WSD) Campaign were essential in transmitting relevant stroke education to the audience in Nepal as depicted by a high engagement of a quarter million social media users in Nepal (n = 265,095). Engagement rates of 1–2% are in general considered a successful social media strategy [30] and our strategy using locally adapted WSD Campaign material achieved an engagement rate of 3.5%, indicating the strong outreach of our awareness program within the target population in Nepal.

Compared to print or mass media campaigns the time for campaign conception and implementation is short, as time consuming processes such as printing, and dissemination are not necessary [31]. Our paid advertisement campaign required only ten working hours of a marketing specialist to engage 2.5 million people, which is in line with other studies highlighting the benefit of digital approaches compared to traditional dissemination strategies [32].

Our sample size was limited by the a priori defined budget for paid advertisement, but even with this limited budget 18% of all social media users in Nepal were exposed to our campaign. We did not observe a decrease in engagement within the time course which leads to the

assumption that we can potentially reach an even larger proportion of the general population if increasing the budget. Furthermore, social media platform analytic tools provide researchers the opportunity to instantly analyze the acceptance of the campaign and to constantly improve the campaign by testing different creatives. Overall, we found that social media is a feasible strategy to disseminate stroke awareness messages to a large audience within a short time-period.

## Cost-effectiveness of social media–based stroke awareness campaigns

The vast majority of impressions (97.8%) and engagements (97.3%) are attributable to paid advertisements and we observed that paid-for advertisements enhanced the number of exposed users by the factor 172 on Facebook and Instagram compared to organic traffic only.

Nonetheless, it is even more remarkable that in Nepal, the cost of social media advertising is far below even the cost of advertising using print media.

During the campaign period of six months, we spent 2,204 EUR for paid advertisements to reach 2.5 million people in Nepal which resulted in a CPC of 0.01 EUR and CPM of 0.24 EUR.

In comparison, a newspaper article in a common Nepali newspaper is estimated to reach 20,000 readers and costs 1,000 EUR [33], resulting in a CPM of 50 EUR, which makes social media 200 times more cost-effective than traditional print media in terms of reaching out to the population. Estimated print media costs to expose one user are even five times higher (0.05 EUR) than the costs for paid advertisement on social media to actively engage (not only to expose) one user (0.01 EUR). Moreover, the costs to produce material for organic traffic on TikTok exceeded the cost of paid advertising (CPM 7 EUR vs CPM 0.24 EUR).

Yet, we have to note that CPC and CPM vary enormously between countries and industries (e.g. average Facebook CPC in Japan 1.6US$ vs 0.2 US$ in Indonesia, average Facebook CPC for healthcare is 1.32$) [34]. As our CPC was only 0.01 EUR, we assume that our results might be generalizable for lower middle-income countries, but not to high-income countries. Another reason for the very low CPC in our study might be the high quality of material used, attracting the attention of social media users. We did not investigate if higher engagement rates and lower costs per click could have been achieved due to tailoring the content to specific target (sub-)audiences.

Overall, paid-for advertising on social media may be a promising health education approach especially for lower middle-income countries and requires further assessment in international trials.

## Findings in comparison with other analyses

Promising results in terms of high impressions on social media were observed in many public health studies [35, 36], however, stroke awareness campaigns have barely used social media so far. While the WSD Campaign provides comprehensive material for social media use (https://www.world-stroke.org/world-stroke-day-campaign), a recent systematic literature review [37] revealed that none of the thirteen identified stroke education studies has evaluated a social media–based approach.

In comparison with other health education studies using paid advertisement on social media, our study resulted in a high engagement rate of 3.5%. A smoking cessation campaign in Egypt led to 17 million impressions for a cost of 5,000 US$, but gained only 13,832 clicks [38] resulting in a > 35 higher cost per click ratio. We mainly attribute our high engagement rate to the appealing visuals of the WSD campaign as affirmative and sentimental content attracts more attention than fear-based approaches [39]. Costs per impression were lower than in other social-media campaigns for health education (also these were run earlier with overall

lower prizes for advertisements). As these campaigns did not analyze temporal trends in advertisement costs we could not conclude if the higher cost-effectiveness of our campaign is a result of the cost optimization of a marketing expert.

Especially taking into account the high costs of mass media campaigns [40] and the high incidence of stroke worldwide, we postulate that social media can be the focus of stroke awareness campaigns as a more cost-effective and measurable approach to reach a larger target population. The impact of such programs on measurable stroke knowledge is unclear, and requires follow-up studies addressing this question.

## Strengths and limitations

This is the first study demonstrating the feasibility of social media as a cost-effective measure for population engagement for stroke awareness in a low-and middle-income country such as Nepal. However, several strengths and limitations should be considered in interpreting the results of our analysis.

First, this analysis focuses on quantitative metrics without assessing the campaign's impact on stroke knowledge. It has been shown elsewhere that being exposed to stroke awareness campaigns can improve knowledge [41, 42], but in this study we did not investigate the association between quantitative metrics and increased stroke knowledge–an important performance indicator [25], which was assessed in a (separately published) case-control study. Furthermore, our study design does not allow analysis on the influence of campaign exposure and assumed increase in stroke awareness on change of behavior in terms of adequate help-seeking behavior. It has previously been shown that increased stroke awareness results in more adequate behavior [42], but this analysis is not provided in our current study.

Second, our analysis indicates the superior cost-effectiveness of our social-media campaign compared to print and mass media. However, our results do not allow conclusions on the value of and audience's attitude towards the campaign. This issue is going to be addressed by a qualitative survey on perception of social media-based stroke education.

Third, the social media-based approach is inevitably accompanied by a selection bias towards literate and affluent people. As our data of the study population is limited to age, gender and place of residence, our study does not allow further conclusions on the distribution of different strata of society reached. Based on the age distribution of Facebook users in Nepal [21], 70% of the exposed users were under the age of 35, which was in line with our target population of young adults as first responders.

Another minor limitation to be mentioned is the limited trackability of organic traffic and its overlap with the paid advertisement audience. While paid traffic was specified to only people residing in Nepal, approximately 10% of the social media users addressed by organic traffic were outside Nepal, totaling less than 0.3% of all engagements. As the social media environment is fragmented across different platforms, this can result in overestimation of the organically reached population by multiple countings. As less than 2% of the study population was reached organically, the impact on our study results is negligible in terms of the overall result. We used paid advertisement for Facebook and Instagram only, but our study excluded other relevant advertisement channels as YouTube, Google and TikTok as well as paid advertisement on Twitter or TikTok, which impedes a comparison between different channels.

Moreover, our campaign focused on stroke awareness material only, and we cannot assume that the results are equally transferable to other relevant health topics. Generalizability of the feasibility and cost-effectiveness of social media-based campaigns to other LMICs and other health topics requires further investigation.

Furthermore, most of the campaign material was provided for free by World Stroke Organisation. Creating (and testing) own materials would have increased the cost of the campaign and time required its implementation immensely. While we assume the quality of the material had a positive effect on the engagement of users, the user's experience should be addressed in a qualitative survey.

Despite these limitations, our study demonstrates that a social media-based campaign with paid advertisement has the potential to inform large parts of the population in an LMIC about stroke with limited monetary and personal resources.

### Future research

The results of our small study indicate the need for and possible effectiveness of further larger-scale stroke education campaigns in Nepal and other countries. A currently ongoing analysis of the campaigns' impact on stroke knowledge is expected to highlight the relevance of social media as a health education channel. Future research should further focus on identification of materials with the highest acceptance within the target population and impact on knowledge.

### Conclusion

Paid advertisements on social media are a cost-effective approach to expose the broad population to stroke awareness information in Nepal. 2.5 million people were reached with a with a high engagement rate and a low Cost-Per-Click of 0.01 EUR. In comparison, organic social media was found to be time consuming and does not achieve a comparable audience within a short time. Future studies are required which address whether exposure to stroke awareness advertisements leads to a measurable increase in knowledge of the warning signs of strokes and necessary subsequent actions.

### Author Contributions

**Conceptualization:** Christine Tunkl, Lekhjung Thapa, Patrick Tunkl, Sarah Belson, Sunanjay Bajaj.

**Data curation:** Patrick Tunkl, Sunanjay Bajaj.

**Formal analysis:** Christine Tunkl, Patrick Tunkl, Sunanjay Bajaj.

**Investigation:** Christine Tunkl, Avinash Chandra, Sunanjay Bajaj, Jessica Golenia.

**Methodology:** Raju Paudel, Patrick Tunkl, Sunanjay Bajaj, Christoph Gumbinger.

**Project administration:** Christine Tunkl.

**Resources:** Patrick Tunkl, Avinash Chandra, Sarah Belson, Nima Haji-Begli, Sunanjay Bajaj, Jessica Golenia.

**Software:** Patrick Tunkl.

**Supervision:** Wolfgang Wick, Werner Hacke, Christoph Gumbinger.

**Writing – original draft:** Christine Tunkl.

**Writing – review & editing:** Patrick Tunkl, Pankaj Jalan, Sarah Belson, Bikram Prasad Gajurel, Sunanjay Bajaj, Christoph Gumbinger.

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
