## [Decision Letter · Decision Letter 0]

16 Mar 2023

PONE-D-23-01201Are digital social media campaigns the key to raise stroke awareness in low-and middle-income countries? A study of feasibility and cost-effectiveness in Nepal.PLOS ONE

Dear Dr. Tunkl,

Thank you for submitting your manuscript to PLOS ONE. After careful consideration, we feel that it has merit but does not fully meet PLOS ONE’s publication criteria as it currently stands. Therefore, we invite you to submit a revised version of the manuscript that addresses the points raised during the review process. Please submit your revised manuscript by Apr 30 2023 11:59PM. If you will need more time than this to complete your revisions, please reply to this message or contact the journal office at plosone@plos.org. Please include the following items when submitting your revised manuscript:A rebuttal letter that responds to each point raised by the academic editor and reviewer(s). You should upload this letter as a separate file labeled 'Response to Reviewers'.A marked-up copy of your manuscript that highlights changes made to the original version. You should upload this as a separate file labeled 'Revised Manuscript with Track Changes'.An unmarked version of your revised paper without tracked changes. You should upload this as a separate file labeled 'Manuscript'.

We look forward to receiving your revised manuscript.

Kind regards,

Ahmed Nasreldein, MD

Academic Editor

PLOS ONE

Journal Requirements:

"This study is supported by Hospital Partnerships funding program of the Deutsche Gesellschaft für internationale Zusammenarbeit (GIZ) GmbH and received funding by the Federal Ministry of Economic Cooperation and Development (BMZ) and the Else Kröner-Fresenius foundation (EKFS)."

"The authors certify that they have no affiliations with or involvement in any organization or entity with any financial interest in the subject matter or materials discussed in this manuscript."

5. We note that Figures 1 and 2 includes an image of a [patient / participant / in the study]. 

Reviewers' comments:

Reviewer's Responses to Questions

**Comments to the Author**

1. Is the manuscript technically sound, and do the data support the conclusions?

Reviewer #1: Yes

Reviewer #2: Yes

2. Has the statistical analysis been performed appropriately and rigorously? 

Reviewer #1: Yes

Reviewer #2: I Don't Know

3. Have the authors made all data underlying the findings in their manuscript fully available?

Reviewer #1: No

Reviewer #2: Yes

4. Is the manuscript presented in an intelligible fashion and written in standard English?

Reviewer #1: Yes

Reviewer #2: Yes

5. Review Comments to the Author

Reviewer #1: Tunkl, et al report the results of a social media stroke awareness campaign in Nepal. They found that paid advertisements resulted in markedly more views/engagements than did organic traffic, and they found that the cost per 1000 views and cost per click were very low (<<<1 EUR). They conclude that social media campaigns are a feasible, cost-effective approach to stroke awareness campaigns in LMICs with the major limitation being that their study did not address whether the campaign was effective at increasing stroke knowledge or optimizing stroke-related behaviors. In general, this is a well-written manuscript of an important, creative and well-designed study. I have a few suggestions to further strengthen the results:

ABSTRACT

1.) Minor point - in the second sentence of the background, it states that lack of stroke awareness hinders the benefits of stroke treatment. I would suggest that lack of awareness does not impact the magnitude of benefit of the treatment. Rather, it hinders the effective provision of stroke treatment (meaning it reaches less people, not that it's less effective for the patients it reaches).

INTRODUCTION

1.) In the 2nd paragraph, it would be helpful to provide a bit of information about the studies that "allow one to assume" that stroke knowledge is low in Nepal.

2.) The authors note that traditional mass media campaigns are often ineffective at improving health knowledge or behavior. Yet, they state that their hypothesis that a social media campaign would be more cost effective than a classical mass media campaign. I would expect that part of the reason for investigating social media campaigns is that they may also be more effective, but this is not included in the author's hypotheses. The authors also allude to reasons why it seems they think a social media campaign may be more effective (targets younger people more and younger people may be more impacted by health education campaigns, etc.). It would make the Introduction stronger if the authors specifically stated any hypotheses as to why a social media campaign may be effective compared to classical mass media campaigns that have been previously shown to have limited effectiveness.

RESULTS

1.) It would be interesting to note whether there were any trends in increased organic traffic to your social media accounts during or right after the paid advertisements?

DISCUSSION

1.) Page 17, line 319 "saturation of engagement" - please define

2.) Page 19, line 340: Is this information on cost in the results section? I can't find it there. If not, would first introduce this in the Results section such that you are not introducing new data in the Discussion section.

3.) In the limitations section, the authors should address the limitations of their hypothesis that their social media campaign was so effective because of the "high quality" and "visually appealing" content. This may be, but users' experience was not assessed so there is no way to know this for certain. Depending on the structure of the planned qualitative survey, this could also be addressed in that work.

4.) Minor error, page 21, line 408 - in the limitations section, it is written "but we CAN assume that the results are equally transferrable to other relevant health topics." Based on the remainder of the paragraph, I believe the authors meant to say "cannot"

DATA POLICY

Of note, the authors do not have a statement regarding data availability within the manuscript.

Reviewer #2: Its an interesting paper which looked at the feasibility and reach of social media (Facebook, Instagram, Twitter, TikTok) for stroke awareness campaigns in Nepal including the cost-benefit analysis.

I looked at twitter for the mentioned handle "@nepalstrokeproject" - but there is no such handle. Instead there is "@NepalStroke" handle which is named "Nepal Stroke Project". This account has 83 followers and it follows 139 people/accounts. The accounts earliest post is on 1/2/2023. Initial tweets had less than 10 likes and got first 10 likes in April 2022 and next >10 like post was the photograph of WSC on 28/10/22. I am not able to verify the views of earlier posts but the recent posts have around 200-300 views.

The authors mention that there is no option of knowing "reach" in twitter - but there is an option of knowing "views" (times a tweet was seen on twitter)

At least in twitter- I couldn't see any major campaign slides or videos from the "@NepalStroke" handle.

In the figure, the authors mention paid campaigns in May and June- but I am not able to see any post in May. Usually in twitter, it will show "promoted" for a paid campaign- but I am not able to see along with any posts.

I don't use Facebook or Instagram much- I think the paper must be reviewed by someone who used FB/Insta for academic purpose.

Overall, the concept is interesting and social media is the future of campaign strategies. The authors should clarify the twitter data if they chose to submit revision.

6. PLOS authors have the option to publish the peer review history of their article (what does this mean?). If published, this will include your full peer review and any attached files.

Reviewer #1: **Yes: **Deanna Saylor, MD, MHS

Reviewer #2: **Yes: **Venugopalan Y Vishnu

---

## [Author Response · Author response to Decision Letter 0]

27 Jun 2023

Reviewer 1: 

1. The abstract was modified according to your suggestions (p.2, line 26: "...the lack of public stroke awareness especially in low- and middle-income countries (LMICs) such as Nepal severely hinders the effective provision of stroke care."

2. We appreciate your feedback on the introduction and elaborated the section about prior studies in Nepal (p.4, ll. 56: "In Nepal, recent studies demonstrated that knowledge on stroke in high-school students and in a rural population is insufficient[7, 8] with e.g. 55% of study participants believing in ayurvedic therapy to be effective in stroke care. The lack of awareness regarding stroke treatments and inadequate recognition of stroke symptoms have been identified as main factors contributing to delayed presentation at a tertiary care center[9] and therefore hampering the benefit of treatment.")

3. We thank you for your comment. Accordingly, we modified the introduction and highlighted the assumed benefits of social media-based campaigns (p.4, ll. 72: "In the last decade, social media has emerged as a platform with an enormous outreach with more than five billion people worldwide owning a smartphone[14]. By facilitating information sharing opportunities and community-building, social media has become invaluable in marketing and a promising approach in health education[15]. Especially young adults have expressed their interest in receiving health information via social media platforms[16] and a review found that social media even has the ability to facilitate mass communication, health education and knowledge translation in LMICs.")

4. We completely agree with you, that the association between paid advertisements and engagement to organic posts would be interesting to study. As our study design intended parallel playout of organic and paid-for posts, our data do not allow to answer your questions. 

5. We reformulated the sentence to improve understandability: p.18, l. 340: “We did not observe a decrease in engagement within the time course which leads to the assumption that we can potentially reach an even larger proportion of the general population if increasing the budget.”

6. We agree with your comment and added the information in the results section (p. 15, l. 303: "TikTok required its own short video format, the costs for each video production were 20 EUR, which adds to a total of 1,130 EUR and a CPM of 7 EUR.")

7. We put into perspective that the user’s experience yet needs to be assessed (p.22: ll. 440: "While we assume the quality of the material had a positive effect on the engagement of users, the user’s experience should be addressed in a qualitative survey.")

8. Thank you for reading carefully, the mistake was corrected (p. 22, l. 434: "... we cannot assume that the results are equally transferable to other relevant health topics"). 

REVIEWER 2: 

- We corrected the Twitter Handle in the manuscript, which is @NepalStroke (p. 7, l. 145). 

- We clarified in the manuscript, that Twitter was used for organic traffic only in contrast to paid-for advertisements on Facebook and Instagram. Further, we highlighted that Twitter was used solely for project-bound information. As correctly stated by the reviewer, Twitter did not reach high numbers of views and engagements, which we attribute to the fact that Twitter was used only for project-specific updates but not for promoting the World Stroke Day Campaign. (p. 7, ll. 148: "While Facebook and Instragram were used to promote the World Stroke Day Campaign, Twitter was used exclusively for promoting project-specific information."; p.9, ll. 193: A predefined action plan combining organic traffic for Facebook, Instagram, Twitter and TikTok and paid-for advertisements on Facebook and Instagram was developed by a marketing expert."). 

- We emphasized in the “limitations” section that Twitter was not comparable to Facebook and Instagram in terms of reach/ engagement, as posts were not promoted with paid advertisements (p.22, ll. 431: "We used paid advertisement for Facebook and Instagram only, but our study excluded other relevant advertisement channels as YouTube, Google and TikTok as well as paid advertisement on Twitter or TikTok, which impedes a comparison between different channels.")

- As the reviewer correctly stated Twitter provides the number of “views”, which we mentioned in the paper as “impressions” according to Twitter’s definition of “view”. As Twitter counts the number of times a post is displayed, we can’t conclude that it was displayed to different people, which is the definition of “reach”.

---

## [Decision Letter · Decision Letter 1]

29 Aug 2023

Are digital social media campaigns the key to raise stroke awareness in low-and middle-income countries? A study of feasibility and cost-effectiveness in Nepal.

PONE-D-23-01201R1

Dear Dr. Tunkl,

We’re pleased to inform you that your manuscript has been judged scientifically suitable for publication and will be formally accepted for publication once it meets all outstanding technical requirements.

Kind regards,

Ahmed Nasreldein, MD

Academic Editor

PLOS ONE

Additional Editor Comments (optional):

Reviewers' comments:

Reviewer's Responses to Questions

**Comments to the Author**

1. If the authors have adequately addressed your comments raised in a previous round of review and you feel that this manuscript is now acceptable for publication, you may indicate that here to bypass the “Comments to the Author” section, enter your conflict of interest statement in the “Confidential to Editor” section, and submit your "Accept" recommendation.

Reviewer #1: All comments have been addressed

Reviewer #2: All comments have been addressed

2. Is the manuscript technically sound, and do the data support the conclusions?

Reviewer #1: (No Response)

Reviewer #2: Yes

3. Has the statistical analysis been performed appropriately and rigorously? 

Reviewer #1: (No Response)

Reviewer #2: Yes

4. Have the authors made all data underlying the findings in their manuscript fully available?

Reviewer #1: (No Response)

Reviewer #2: Yes

5. Is the manuscript presented in an intelligible fashion and written in standard English?

Reviewer #1: (No Response)

Reviewer #2: Yes

6. Review Comments to the Author

Reviewer #1: (No Response)

Reviewer #2: All reviewer comments regarding this paper have been addressed adequately by the authors. I have no more comments

7. PLOS authors have the option to publish the peer review history of their article (what does this mean?). If published, this will include your full peer review and any attached files.

Reviewer #1: **Yes: **Deanna Saylor

Reviewer #2: **Yes: **Venugopalan Y Vishnu

---

## [Editor Report · Acceptance letter]

31 Aug 2023

PONE-D-23-01201R1 

Are digital social media campaigns the key to raise stroke awareness in low-and middle-income countries? A study of feasibility and cost-effectiveness in Nepal. 

Dear Dr. Tunkl:

I'm pleased to inform you that your manuscript has been deemed suitable for publication in PLOS ONE. Congratulations! Your manuscript is now with our production department. 

Kind regards, 

on behalf of

Dr. Ahmed Nasreldein 

Academic Editor

PLOS ONE